# Taxifolin Alleviates DSS-Induced Ulcerative Colitis by Acting on Gut Microbiome to Produce Butyric Acid

**DOI:** 10.3390/nu14051069

**Published:** 2022-03-03

**Authors:** Wei Li, Le Zhang, Qingbiao Xu, Wenbo Yang, Jianan Zhao, Ying Ren, Zhendong Yu, Libao Ma

**Affiliations:** College of Animal Science and Technology, Huazhong Agricultural University, Wuhan 430070, China; hzau1898@gmail.com (W.L.); hzau2022@163.com (L.Z.); qbxu@mail.hzau.edu.cn (Q.X.); yangwenbo@webmail.hzau.edu.cn (W.Y.); zjn392611@163.com (J.Z.); hzau2023@163.com (Y.R.); zdyhzau@163.com (Z.Y.)

**Keywords:** fecal microbiota transplantation, butyric acid, ulcerative colitis

## Abstract

Taxifolin is a bioflavonoid which has been used to treat Inflammatory Bowel Disease. However, taxifolin on DSS-induced colitis and gut health is still unclear. Here, we studied the effect of taxifolin on DSS-induced intestinal mucositis in mice. We measured the degree of intestinal mucosal injury and inflammatory response in DSS treated mice with or without taxifolin administration and studied the changes of fecal metabolites and intestinal microflora using 16S rRNA. The mechanism was further explored by fecal microbiota transplantation. The results showed that the weight loss and diarrhea score of the mice treated with taxifolin decreased in DSS-induced mice and longer colon length was displayed after taxifolin supplementation. Meanwhile, the expression of GPR41 and GPR43 in the colon was significantly increased by taxifolin treatment. Moreover, the expression of TNF-α, IL-1β, and IL-6 in colon tissue was inhibited by taxifolin treatment. The fecal metabolism pattern changed significantly after DSS treatment, which was reversed by taxifolin treatment. Importantly, taxifolin significantly increased the levels of butyric acid and isobutyric acid in the feces of DSS-treated mice. In terms of gut flora, taxifolin reversed the changes of *Akkermansia*, and further decreased *uncultured_bacterium_f_Muribaculaceae*. Fecal transplantation from taxifolin-treated mice showed a lower diarrhea score, reduced inflammatory response in the colon, and reduced intestinal mucosal damage, which may be related to the increased level of butyric acid in fecal metabolites. In conclusion, this study provides evidence that taxifolin can ameliorate DSS-induced colitis by altering gut microbiota to increase the production of SCFAs.

## 1. Introduction

Inflammatory bowel disease (IBD) is a kind of gut disorder whose etiology has not been fully elucidated, consisting of ulcerative colitis (UC) and Crohn’s disease [1]. UC was first described in 1859 and was characterized by mucosal inflammation that begins in the rectum and the proximal to the colon [2,3,4,5]. In recent years, UC has become a chronic disease worldwide [6]. The causes of UC are complex and unclear. The clinical manifestations of UC are mainly rectal bleeding, diarrhea, tenesmus, and sometimes low abdominal pain [7]. Previous studies have found that the UC recurred in about 15% of patients five years after diagnosis, and in up to about 25% of patients at 10 years [8]. Therefore, the establishment of animal models similar to humans is of great significance for studying the pathogenesis of UC and predicting the clinical efficiency of UC.

There are trillions of microorganisms in the animal gastrointestinal tract [9]. A growing amount of research evidence shows that differences in intestinal flora have a vital role in colitis [10,11,12]. Many clinical studies have confirmed that UC is related to intestinal microbiota disorder, where the composition of flora changes and some strains increase [12,13]. The characteristic changes of UC flora were explored and found that the intestinal flora of UC patients was characterized by the decrease of Firmicutes, the increase of proteobacteria, and the decrease of butyric-producing bacteria (e.g., *Roseburias* and *Faecalibacterium*) [13,14]. In addition, previous studies have suggested oral probiotics can prevent DSS-induced intestinal mucositis [15,16]. Moreover, DSS-induced intestinal mucositis attenuated after fecal transplantation in healthy mice, suggesting that intestinal microbiota is actively involved in DSS-induced intestinal pathology [17]. However, effective drugs are still lacking to relieve gut injury caused by DSS. Previous studies have shown that adding plant extracts had significant therapeutic effects on mice with liver cancer, colitis, kidney disease, and other inflammatory diseases [18]; research suggests that it works in part by altering the microbiota structure of gut microbes. 

Taxifolin, also known as dihydroquercetin, is a bioflavonoid and quasi-vitamin P [19]. Taxifolin is a naturally active product that exhibits anti-inflammatory, antioxidant, antibacterial, and other biological activity [20,21,22]. Moreover, it is non-toxic, non-teratogenic, and non-mutagenic [22,23]. In recent years, the medicinal properties of taxifolin have become a research hotspot in pharmaceutics. Taxifolin can remove free radicals and toxins effectively that have destructive functions in the human body, protect and repair cells, promote the absorption of vitamin C, and prevent vitamin C from being oxidized. Studies have shown that taxifolin reduces free radicals produced by mitochondria in cells and exerts an antioxidant effect by reducing the activity of xanthine oxidase [23,24]. Ahn et al. also found that taxifolin can reduce the release of inflammatory cells, which can be used to treat atopic dermatitis [25]. In addition, an increasing amount of research has shown that taxifolin regulates intestinal flora in the colon [26]. For example, taxifolin can treat metabolic disorders induced by obesity in rats via modulating intestinal flora [27]. Moreover, it was suggested that inflammation played a critical role in the pathological process of UC, and taxifolin protected DSS-induced UC by inhibiting the inflammatory response [26]. Recently, taxifolin was also confirmed to be a regulator by modifying the intestinal flora of mice [27]. In addition, short-chain fatty acids (SCFAs) can reduce inflammation, and polyphenols may regulate the intestinal microbial ecosystem and promote the production of SCFAs by intestinal bacteria [28].

In conclusion, the therapeutic mechanism of taxifolin on DSS-induced intestinal mucositis remains unclear. We speculate that taxifolin may act through gut microbes. In this study, the effects of taxifolin on intestinal mucositis induced by DSS were investigated by observing the inflammatory response, SCFAs and intestinal flora. In combination with fecal microbiota transplantation (FMT), we found that taxifolin regulated the SCFAs, inhibited the expression of TNF-α, IL-1β, and IL-6 in the colon, and alleviated DSS-induced colitis by regulating the gut microbiome in mice. 

## 2. Materials and Methods

### 2.1. Animal Experiments

Female, 8-week-old C57BL/6 mice were obtained from the Laboratory Animal Center of Huazhong Agricultural University (Wuhan, China). The mice lived under a light/dark cycle (26 ± 2 °C, 55 ± 10% relative humidity) for 12 h with free access to food and water. After 3 days of adaptive feeding, mice were randomly divided into 3 groups (*n* = 7/group): control group, DSS group, and DSS + Taxifolin group. Experimental colitis was induced by replacing the drinking water with 5% DSS (Sangon Biotech (Shanghai) Co., Ltd., Shanghai, China) for 7 days (from day 1 to day 7) [29]. While mice in the control group were orally administered ddH_2_O. Meanwhile, mice in the DSS + Taxifolin group were orally administered 150 μL Taxifolin (200 mg/kg in ddH_2_O) daily for 7 consecutive days (days 8–14). Mice in the control and DSS groups were given 150 μL ddH_2_O (10 mL/kg) per day. All animal experiments were performed in accordance with institutional animal care guidelines approved by the Laboratory Animal Ethics Committee of Huazhong Agricultural University.

### 2.2. Fecal Microbiota Transplantation

For FMT experiment, 14 mice were randomly divided into two groups: the FMT-Control (FC) group and the FMT-Taxifolin (FT) group (*n* = 7/group). Experimental colitis was induced by replacing the drinking water with 5% DSS for 7 days (from day 1 to day 7). While mice in the control group were orally administered given ddH2O. Fresh feces for murine FMT were shipped on ice to an anaerobic incubator in the laboratory within 30 min of defecation, and 200 mg of mixed feces were diluted with 2 mL of sterile saline and the fecal bacteria solution was gently mixed using a homogenizer,. The homogenate is then passed through a filter to remove small undigested particles from the fecal suspension, and 150 μL (2 × 10^8^−2 × 10^9^ cfu/mL) bacterial suspension was used for gavage into each mouse. 

FC mice were orally gavaged with fecal suspension from the control group once a day for 7 days (days 8–14). Meanwhile, mice in the FT group were orally gavaged with the fecal suspension of the DSS + Taxifolin group once a day for 7 days. The mice were not pretreated with antibiotics considering that antibiotic pretreatment has an effect on the efficacy of fecal transplantation [30] and may affect intestinal mucositis [31]. The body weight and diarrhea of the mice were recorded daily. At the end of the experiment, mice were executed by vertebral dislocation and the length of the induced colon was measured.

### 2.3. Disease Activity Index

Includes 3 indicators: weight loss (0–4, 0 to 20% loss), stool consistency (0 for normal, 2 for loose stools, and 4 for diarrhea) and fecal occult blood (0 points for normal, 2 points for positive occult blood, and 4 points for overt hemorrhage).

### 2.4. Histologic Analysis of Mice Colon

Disease severity, including the degree of body weight (BW) reduction and diarrhea severity, was recorded daily to assess colitis. Diarrhea severity was classified into five classes according to fecal consistency: 0, normal; 1, slightly moist; 2, moderately moist; 3, loose; and 4, watery stools [32]. On day 15, all mice were sacrificed. The distal colon (approximately 3.5 cm from the anus) was washed with ice-cold phosphate-buffered saline (PBS), and the length of the colon was measured. The 3 cm ileum and colon tissues near the ileum were fixed in 4% paraformaldehyde, dehydrated in ethanol, and embedded in paraffin. A portion of the ileum and colon samples were snap frozen in liquid nitrogen for further analysis. The morphology of the colon was observed using a light microscope. The degree of tissue inflammation damage was scored according to the degree of inflammatory cell infiltration and the degree of tissue damage.

### 2.5. Gut Microbiota Analysis

We selected fresh feces from 5 mice in each group ((Control, DSS, Taxifolin, FC and FT) for 16S rRNA sequencing and performed 16S rRNA sequencing. Then, microbial genomic DNA was extracted using Fecal genomic DNA extraction kit (DP328) (TIANGEN, Beijing, China) according to the manufacturer’s instructions. The raw data were quality filtered using Trimmomatic (version 0.33), then the primer sequences were identified and removed using Cutadapt2 (version 1.9.1), followed by USEARCH (version 10) to splicing the paired-end reads and removing chimeras (UCHIME, version 8.1), resulting in high-quality sequences for subsequent analysis. Labels with more than 97% similarity are clustered into an operational taxonomic unit (OTU). OTU representative sequences were analyzed using the RDP Classifier Bayesian algorithm, and community composition was analyzed at the phylum, class, order, genus, and species levels.

### 2.6. SCFA Quantification

After the mice were sacrificed, Colon contents were collected. All samples were immediately frozen on dry ice. The standards of acetic acid, propionic acid and butyric acid were weighed and mixed in a 10 mL flask to prepare a gradient mixture, and the standard curve was calculated and plotted according to the chromatogram of the corresponding concentrations [33]. Weigh a certain amount of mice feces, add 2 mL of ultrapure water at 1 g, mix thoroughly and then centrifuge at 4 °C for 10 min at 1000 rmin^−1^, extract the supernatant through 0.22 μm filter membrane into gas chromatography, and calculate the concentrations of acetic acid, propionic acid and butyric acid.

### 2.7. Total RNA Extraction and Real-Time Quantitative PCR (qPCR) 

Individual colon tissue samples (10–20 mg) were homogenized using a Tgrinder Electric Tissue Grinder (TIANGEN, Beijing, China) with a mortar and pestle and buffer RL (TIANGEN, Beijing, China). Total RNA was isolated according to the manufacturer’s instructions (RNA prep Pure Tissue Kit, TIANGEN, Beijing, China). The primer sequence for quantitative real time PCR is shown in Appendix A. Complementary DNA (cDNA) was prepared from 1 μg of total RNA using the Prime Script TM RT kit and DNA Eraser. The reaction was performed in a 20 μL system: 2 μL of 5 ng/uL template, 3 μL of each upstream and downstream primers at 10 μumoL/L, 10 μ of 2xRealStar Green Fast Mixture, and 20 μL of deionized water. The fluorescence signal was collected at the end of the reaction at 72 °C, and the lysis curve was set up from 65 to 95 °C with 0.5 °C/5 s increments. The results were evaluated based on the exponential growth of fluorescence signal, quantification cycle (Cq) values and dissolution curves. The data were analyzed using the 2^−ΔΔCt^ method, where the β-action was used as an endogenous control gene. 

### 2.8. Statistical Analysis

Statistical analysis was performed using GraphPad Prism 8.0 software, data were expressed as mean ± standard deviation (*x* ± s), and comparisons between groups were performed by one-way ANOVA and LSD test. *p* < 0.05 indicated that the difference was statistically significant.

## 3. Results

### 3.1. Effects of Taxifolin and FMT on Mice Colitis

Cumulative studies showed FMT can treat gut mucositis by altering intestinal microbiota [32,34,35]. Previous studies have shown some effect of Taxifolin on DSS-induced intestinal mucositis in mice. The design of this study was shown in Figure 1A, and the intestinal flora of the control group and DSS + Taxifolin group were transferred to DSS-induced mice colitis group. The results were consistent with previous studies [26]. The body weight (ΔBW) of the control group increased by 0.32 ± 0.01 g. While ΔBW in the DSS group decreased 3.54 ± 0.07 g after DSS treatment (*p* < 0.05) (Figure 1B). In contrast, ΔBW in the DSS + Taxifolin group decreased by 2.75 ± 0.07 g, which was significantly lower than that that in the DSS group (*p* < 0.05) (Figure 1B). In addition, we found that ΔBW in the FC and FT groups decreased by 1.86 ± 0.11 g and 1.63 ± 0.14 g, respectively, which was also significantly lower than the average BW loss of the DSS group (*p* < 0.05) (Figure 1B). The diarrhea score in the DSS + Taxifolin group was significantly lower than that in the DSS group. Consistent with the DSS + Taxifolin group, diarrhea was significantly reduced in the FC group and the FT group at day 13 compared to the DSS group (*p* < 0.05) (Figure 1C). The mice colon length was significantly decreased by DSS treatment (Figure 1D, *p* < 0.05), and FMT treatments significantly rescued this loss (*p* < 0.05) (Figure 1D). The DAI index of the T group, the FC group and the FT group was significantly lower than that of the DSS group after the 9th day, that is, the diarrhea, blood in the stool and weight loss were relieved. The T group had the same trend as the FC group and the FT group (*p* < 0.05) (Figure 1E). 

As reported in previous studies [26], the intestinal mucosa of DSS-treated mice was significantly damaged. DSS treatment damaged crypt-villus structures and increased inflammatory cell infiltration (Figure 2A,B). In contrast, taxifolin reduced mucosal damage and inflammatory cell infiltration (Figure 2A,B). In addition, fecal transplantation from DSS + Taxifolin group and fecal transplantation from Control group alleviated mucosa atrophy, villi loss, and the degeneration and necrosis of colon epithelial cells caused by DSS treatment (Figure 2A,B). 

### 3.2. Taxifolin and FMT Recovered SCFA Content in C57BL/6 Mice

Based on previous studies, adding Taxifolin can regulate the metabolisms of alanine, aspartate, glutamate, methane, amino sugar, and nucleotide sugar [36]. Lots of polyphenols can increase the amount of the bacteria that can produce SCFAs [37,38,39], which is associated with intestinal mucositis and gut microbiota [40,41]. Previous studies have shown that the SCFA-mediated activation of GPR41/GPR43 pathways and their inhibition on histone deacetylases have been widely studied [42,43]. Therefore, we detected the mRNA expression of G-protein coupled receptor 41 (GPR41) and G-protein coupled receptor 43 GPR43 in colon tissues to reflect the content of SCFAs in vivo. A decrease in the levels of (GPR41) and (GPR43) were found by DSS treatment. After Taxifolin treatment, the level of GPR41 and GPR43 were increased significantly (Figure 3A). In addition, the expression of GPR41 and GPR43 mRNA was increased in the FT and FC groups compared with the DSS group. For fecal SCFAs levels, Taxifolin treatment significantly increased propanoic acid, butyric acid and isobutyric acid production, and the FMT(Control) treatment also significantly increased butyric acid and isobutyric acid production, and FT treatment increased isobutyric acid and isovaleric acid production. Pentatonic acid exhibited the same trend, but the difference did not reach significant values (Figure 3C). These results showed that Taxifolin reversed the changes of SCFAs caused by DSS, and the gut microbiota from the control and DSS + Taxifolin groups can also reverse the changes of SCFAs by DSS treatment. 

### 3.3. Taxifolin and FMT Decreased the Levels of Inflammatory Cytokines in C57BL/6 Mice

After the injection of DSS, we observed that the level of IL-1β, IL-6, and TNF-α in colon was significantly increased compared to control group. Interestingly, taxifolin significantly reduced the increase of IL-1β, IL-6, and TNF-α in DSS group, suggesting that taxifolin reduced the inflammatory response in DSS-induced mice. In addition, we examined the expression of inflammatory factors in colon tissue from FC and FT group, and found that the levels of TNF-α, IL-1β, and IL-6 were significantly reduced in FT and FC compared to DSS group (*p* < 0.05) (Figure 4). These results suggested that the intestinal flora mediated by Taxifolin treatment alleviated DSS-induced mice colitis.

### 3.4. Taxifolin and FMT Reduced the Expression of Inflammatory Cytokines via NF-κB Signaling Pathway

To detect whether Taxifolin exerts anti-inflammatory effects through NF-κB signaling pathway, we examine the mRNA expression of NF-κB P65 and IκBα of the mice. After the injection of DSS, a significant increase was observed in the level of NF-κB P65 by DSS treatment. By contrast, the level of IκBα in the colon from DSS group was observed to have a significant decrease compared to the control group. The increase in the level of NF-κB P65 caused by DSS treatment were significantly reduced by Taxifolin treatment and the level of IκBα was significantly increased by Taxifolin treatment, demonstrating that Taxifolin reduced inflammatory response by NF-κB signaling pathways by DSS treatment. Additionally, the levels of NF-κB P65 were significantly decreased in FT and FMT(Control) compared to DSS group (*p* < 0.05), the levels of IκBα were significantly increased in FT and FC compared to DSS group (*p* < 0.05) (Figure 5). These results suggested that the intestinal flora which changed by taxifolin treatment relieved intestinal mucositis induced by DSS treated in mice by the NF-κB signal pathway.

### 3.5. Taxifolin and FMT Alter the Gut Microbiota in DSS-Induced Intestinal Mucositis

Intestinal flora has a critical role in inflammatory bowel disease [44]. DSS altered the composition of intestinal flora, which was associated with the development of colitis. The diversity and community composition of intestinal flora in feces were analyzed by Miseq sequencing. In order to clarify the changes of intestinal flora, PCA showed that the intestinal flora of the DSS group were separated from the control group, and the distance between DSS + Taxifolin and the control group was closer than that between DSS and control group, suggesting that the composition of intestinal flora of C57BL/6 mice treated with taxifolin was closer to the control group. Moreover, a PCA scatter plot showed that both the FC group and FT group samples were separated from the DSS group, and the samples from the FC group was very close to samples from the control group, indicating that the composition of intestinal flora by FMT(Control) was closer to the control group (Figure 6A). Then, β-diversity analysis revealed significant differences in abundance and composition between the five groups at the phylum level (Figure 6B). Compared with control group, treatment with DSS reduced the relative abundance of *Bacteroidetes*. In addition, there were significant differences in abundance and composition between the five groups at the level of class and genus (Figure 6C–F). Compared with the control group, treatment with DSS reduced the relative abundance of *uncultured_bacterium_f_Muribaculaceae* and *Akkermansia*, treatment with taxifolin decreased *uncultured_bacterium_f_Muribaculaceae*, FC and FMT (DSS+ Taxifolin) treatment increased the relative abundance of *Akkermansia* and *Lactobacillus*, but decreased the relative abundance of Bacteroides. These results suggest that both Taxifolin treatment and FMT were able to repair DSS-induced intestinal flora dysbiosis in mice, but the species and abundance of beneficial intestinal flora affected differed.

In conclusion, DSS treatment leads to intestinal flora disorder, increasing the content of harmful bacteria and reducing the content of beneficial bacteria. However, taxifolin treatment significantly increased the abundance of beneficial bacteria and decreased the abundance of harmful bacteria, suggesting that taxifolin recued the disrupted intestinal flora caused by DSS in mice. The trend of the effect of FT group on intestinal flora was consistent with that of T group, further demonstrating that it is Taxifolin that plays a regulatory role on the abundance of intestinal flora.

### 3.6. Predicted Metabolomic Profiles of Microbiota from Control and DSS and Taxifolin and FMT Group

In order to examine whether Taxifolin and FC and FT affected the metabolic activity of microbial communities in the feces, we utilized the PICRUSt algorithm to infer their metagenomes and collapse the genes into the Kyoto Encyclopedia of Genes and Genomes (KEGG) pathway at level 2 (Figure 7). Figure 7 shows that the Carbohydrate metabolism pathway was predicted to be enriched in the Control and DSS + Taxifolin and FC and FT group. These results indicated that FMT may restore the colonic mucosal damage by increasing the Carbohydrate metabolism pathway.

## 4. Discussion

Previous studies have demonstrated that taxifolin relieved DSS-induced colitis by NF-κB signal way [26], and Su et al. reported that taxifolin, which could improve the obesity symptoms, hepatic steatosis, and gut microbiota dysbiosis in HFD fed C57BL/6 mice [27]. However, no relevant studies have reported that taxifolin could relieve intestinal mucositis induced by DSS treatment by regulating intestinal flora. In this study, we detected the alterations of intestinal mucosa, the mRNA expression of inflammatory cytokines, the distribution of short chain fatty acids, the mRNA expression of SCFA receptor and intestinal flora with or without Taxifolin treatment. We found that the content of SCFAs, the mRNA expression of inflammatory cytokines, the mRNA expression of the SCFA receptor and intestinal microflora were significantly altered by taxifolin administration in DSS-induced mice with mucosal barrier destruction. Further research showed that the inflammatory factors and SCFAs also changed significantly after FMT. Moreover, the severity of intestinal mucositis in FMT from DSS + Taxifolin and FMT from the Control group was significantly reduced compared to DSS treated mice. Therefore, our results suggested that taxifolin may alter the content of intestinal SCFAs by regulating intestinal flora, thereby reducing intestinal inflammation and relieving DSS treated intestinal mucositis.

The increased expression of inflammatory factors is closely associated with the occurrence of intestinal mucositis and is related to NF-κB pathways [45]. There is increasing evidence suggesting that anti-inflammation effect could improve UC and inhibit its progression [46,47,48]. Taxifolin, a natural active drug, has been treated for the management of some diseases. Cai et al. reported that taxifolin decreased serum levels of TNF-α, IL-1β and IL-6, and is considered a potential alternative therapeutic agent for osteoclast related diseases [49]. Consistent with previous research [26], the levels of inflammatory factors in colon tissues increased significantly after DSS treatment, indicating that DSS caused intestinal mucositis by enhancing inflammatory cytokine expression. However, the expression of inflammatory factors was decreased by the treatment of taxifolin, suggesting that taxifolin could alleviate inflammatory responses induced by DSS treatment. The fact that the FMT (control) treatment could not reverse the decrease of IL-10 by DSS may be owing to the fact that it alleviates DSS-induced ulcerative colitis by increasing the acetic acid producing bacteria.

Previous studies have shown that metabolites of bacterial flora are related in colitis development [50]. Zhou et al. have reported that SCFAs could alleviate DSS-induced colitis by regulating autophagy via stabilizing HIF-1α. The metabolites SCFAs have a vital role in the protection of intestinal mucosa by reducing the expression of inflammatory cytokines by reducing inflammation levels [51]. In the present study, taxifolin significantly changed the SCFAs receptor expression in DSS treated mice. Taxifolin significantly increased the content of butyric acid and isobutyric acid in colitis mice. Furthermore, among analysis of the Kyoto Encyclopedia of Genes and Genomes (KEGG) metabolic pathways, we focused on the Carbohydrate metabolism known to be associated to gut barrier [52,53,54], and found that taxifolin could improve disorders of glucose metabolism in metabolic syndrome rats [38]. Studies have shown that the metabolite butyrate inhibited colitis by regulating the differentiation of Th1 and Th17 and promoting the production of IL-10 [55]. In addition, sodium butyrate alleviates TNBS-induced inflammatory response by activating GPR109A and inhibiting pro-inflammatory signaling pathways (NF-κB and AKT signaling pathways), promotes gut barrier function repair, increases TJ protein expression, and protects cells from apoptosis in pathological states [56]. These studies showed that SCFAs alleviates intestinal mucositis by reducing inflammatory cytokine expression and enhancing intestinal mucosal barrier, suggesting that taxifolin may reduce intestinal inflammation by increasing the content of SCFAs and increasing butyric acid levels.

The disorder of intestinal flora is associated with the pathological development of IBD [57,58]. Studies have shown that DSS treatment could make intestinal mucositis worse by changing the intestinal flora, and taxifolin could change the composition of intestinal flora [26]. However, there is no direct evidence showed that taxifolin could alleviate DSS caused intestinal mucositis by improving the intestinal flora environment. To investigate whether the intestinal flora is involved in the regulation of taxifolin on DSS-induced intestinal mucositis in mice, we detected the intestinal flora after treatment with DSS and Taxifolin. We found that taxifolin treatment changed the structure of the gut microbiota caused by DSS in mice, and had a reverse effect on the destruction of gut microbiota caused by DSS. In addition, we found that taxifolin increased the abundance of *Firmicutes* and decreased the abundance of *Bacteroidetes* of mice. Low abundance of *Firmicutes* microbiota can increase intestinal sensitivity to inflammation [59,60,61], suggesting that taxifolin could play an anti-inflammatory role in the intestinal tract by regulating intestinal microbiota imbalance. More importantly, a large number of the top 20 genera, 8 beneficial bacteria were significantly enriched after Taxifolin treatment, and 6 of them belong to SCFA-producing bacteria, such as *Ruminiclostridium_5*, *Bacteroides*, *Lachnospiraceae_NK4A136_group*, *Lactobacillus*, *Ruminococcaceae_UCG-014*, [*Eubacterium]_coprostanoligenes_group* [62,63,64,65,66,67]. It has been reported that SCFA-producing bacteria can benefit the host by inhibiting the expression of inflammatory factors, providing nutritional levels to colon cells, and enhancing tight junction protein expression [68]. Instead, enteritis progress can be promoted by pathogenic bacteria. For instence, *Blautia* were positively associated with T2D [69]. Ruminiclostridium_6 has been shown to be involved in enhancing inflammatory responses [70]. In our study, DSS treatment significantly increased the relative abundance of *Blautia* and *Ruminiclostridum_6* in mice, which might be responsible for inducing colitis. Importantly, Taxifolin can decrease the level of *Blautia* and *Ruminiclostridum_6*. Therefore, intestinal mucosa injury caused by DSS is closely related to the intestinal flora disorder caused by it, and taxifolin might alleviate DSS-induced colitis by enriching SCFA-producing bacteria and inhibiting pathogenic bacteria. However, it is not clear whether these bacteria are directly involved in colitis development, in particular whether taxifolin prevented the development of intestinal inflammation a by altering the structure of the intestinal flora to produce different fecal metabolites.

Different intestinal microbiota compositions have different metabolic levels. Studies have shown that the intestinal flora treats gastrointestinal health through its metabolite SCFAs [71]. In obesity, intestinal microbes produced energy-related metabolites to regulating the host’s energy harvest [72]. The gut microbiome can work by affecting its metabolites. FMT is an effective way to study the relationship between drugs, microflora, and metabolites. We found significant differences in fecal metabolites between two groups after FMT, which may be related to differences in intestinal flora composition. In addition, we found an interesting phenomenon in our research, treatment with Taxifolin or treatment with FMT(Control) or FT could all improve intestinal mucosal lesions, and all of these methods improve intestinal inflammatory responses by increasing the proportion of beneficial bacterium. Treatment with Taxifolin increased the abundance of *Ruminiclostridium_5*, *Bacteroides*, *Lachnospiraceae_NK4A136_group*, *Lactobacillus*, *Ruminococcaceae_UCG-014*, *[Eubacterium]_coprostanoligenes_group*. Treatment with FMT(Control) and FMT(Taxifolin) increased the abundance of *uncultured_bacterium_o_Mollicutes_RF39*, *Turicibacter*, *Akkermansia* [38,73,74]. Moreover, KEGG analysis indicated that DSS + Taxifolin, FMT(Control) and FT could all increased the carbohydrate metabolism compared with DSS group. Meanwhile, the level of GPR41 and GPR43 mRNA expression was obviously higher in FT group mice. As shown in Figure 3B, the SCFAs content in fecal in FT and FC group was significantly increased compared with the DSS group. We found that the FC group have a higher content in butyric acid, the FT group have a higher content in isobutyric acid and isovaleric acid compared with the DSS group, which may be influenced by the abundance of *Akkermansia* and *Lactobacillus* and *[Eubacterium]_coprostanoligenes_group* in gut microbiota. All of them can produce butyric acid [75]. More importantly, fecal transplantation from DSS + Taxifolin mice reversed diarrhea caused by DSS treatment, reduced the inflammatory responses in colon tissues, and enhanced the intestinal barrier. These results indicated that the therapeutic effect of Taxifolin on intestinal inflammation were partially realized by regulating the composition of intestinal flora.

Our research results showed that taxifolin can regulate the level of fecal metabolites and increased the level of GPR41 and GPR43 in the colon, and increased the level of the content of SCFAs, especially butyric acid by regulating the intestinal microbiota, thereby reducing DSS-induced intestinal inflammatory reaction and protecting the intestinal mucosa. In summary, this study provides a potential new strategy for the management of Colitis in humans and animals with taxifolin and other traditional Chinese medicine.

## Figures and Tables

**Figure 1 nutrients-14-01069-f001:**
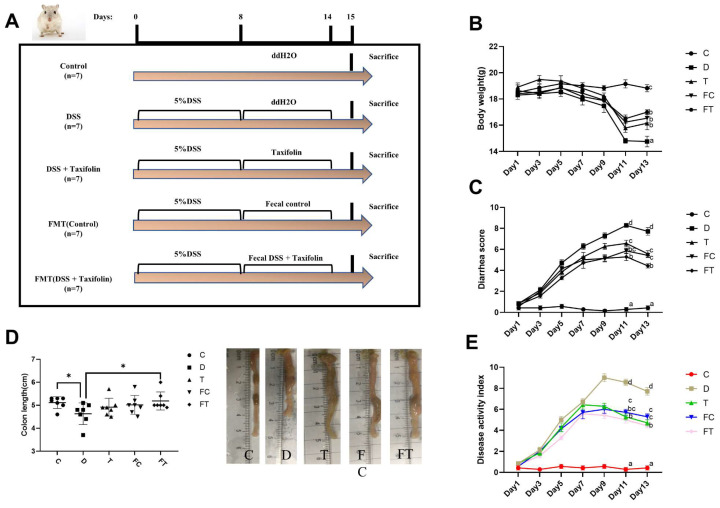
Taxifolin and FMT alleviated DSS-induced intestinal mucositis in mice. (**A**) Experiment design. The DSS group and DSS + Taxifolin and FC (FMT Control) and FT(FMT DSS + Taxifolin) group mice were intraperitoneally administered with DSS (5%) once daily for 7 days (days 1–7). The control group mice were orally administered with ddH2O. Meanwhile, the DSS + Taxifolin group were treated with Taxifolin (200 mg/kg in ddH_2_O) by oral gavage once daily for 7 days (days 8–14) and FC and FT were treated with 150 μL fecal suspension (100 mg/mL) from control and DSS + Taxifolin group respectively. (*n* = 7) (**B**) Body weight loss after DSS administration were significantly repressed after Taxifolin or FMT treatment. (**C**) Mice treated with Taxifolin or FMT had a significantly lower score of diarrheas compared with DSS group, especially in day11 and day13. (**D**) Taxifolin treatment reduced the loss of colon length caused by DSS treatment. (**E**) DAI scores for experimental procedures in mouse. Values were expressed as mean ± SEM (B–D: *n* = 7) * *p* < 0.05, The letters abcd are significant markers, and any difference with one of the same marker letters is not significant, and any difference with different marker letters is significant.

**Figure 2 nutrients-14-01069-f002:**
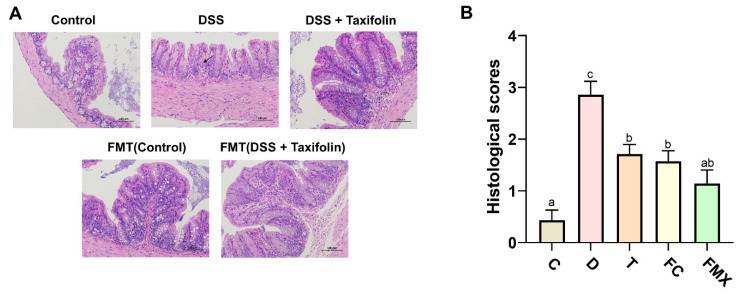
Taxifolin and FMT alleviated DSS-induced intestinal mucositis in mice. H&E staining of colon sections (200 ×). (**A**) Control mice showing normal mucosa; DSS group, showing mucosa atrophy and villi loss with the degeneration and necrosis of epithelium cells and infiltration of inflammatory cells; DSS + Taxifolin group mice, showing mucosa had better morphological structure and infiltrated with less inflammatory cells; FC(FMT Control) mice, showing mucosa had better morphological structure; FT(FMT DSS + Taxifolin) mice, showing mucosa had better morphological structure. (**B**) The histological scores of the five group. The letters abc are significant markers, and any difference with one of the same marker letters is not significant, and any difference with different marker letters is significant.

**Figure 3 nutrients-14-01069-f003:**
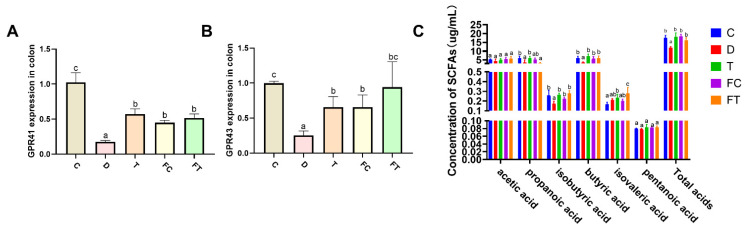
Taxifolin and FMT reserved the expression level of short chain fatty acid (SCFAs) receptor in colon tissue in DSS treated mice. The expression levels of SCFAs receptor GPR41 (**A**), GPR43 (**B**) in each group were detected. Data are mean ± SEM (*n* = 7). (**C**) Test results of SCFAs in feces. Content in stool samples of mice. The letters abc are significant markers, and any difference with one of the same marker letters is not significant, and any difference with different marker letters is significant.

**Figure 4 nutrients-14-01069-f004:**
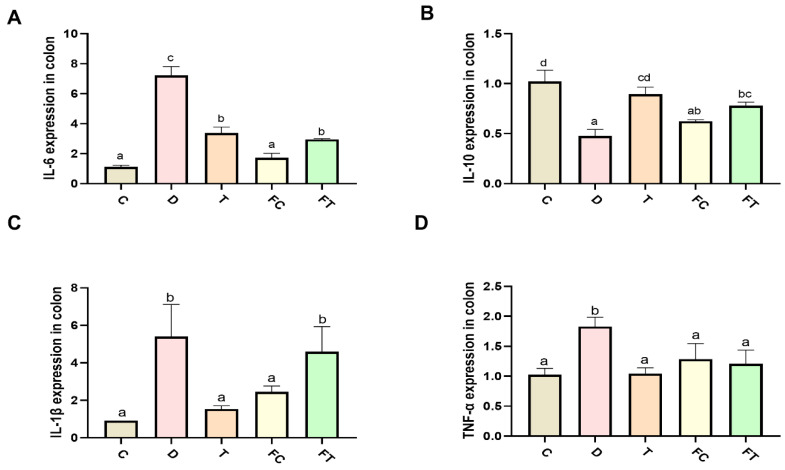
Taxifolin and FMT treatment reduced the expression level of inflammatory cytokines in colon tissue in DSS treated mice. The expression level of inflammatory cytokines IL-6 (**A**), IL-1β (**B**), TNF-α (**C**), IL-10 (**D**) in each group were detected. Data are mean ± SEM (*n* = 7). The letters abcd are significant markers, and any difference with one of the same marker letters is not significant, and any difference with different marker letters is significant.

**Figure 5 nutrients-14-01069-f005:**
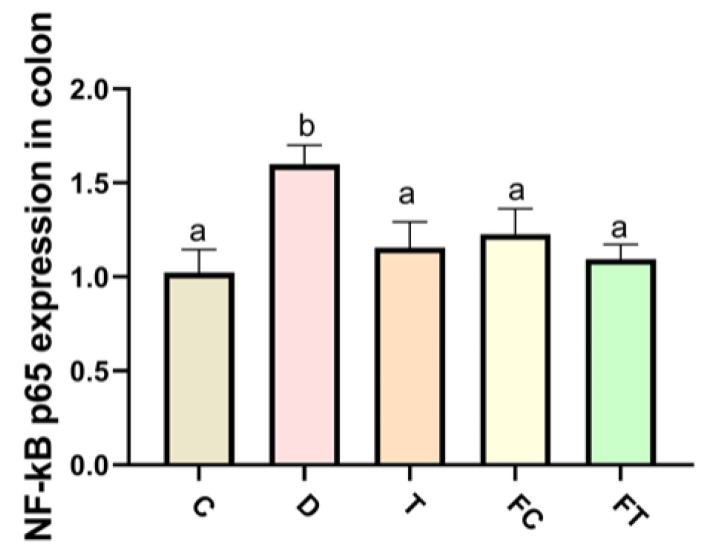
Taxifolin and FMT reduced the expression of inflammatory factors by the NF-κB signal path in colon tissue in DSS treated mice. The expression level of the NF-κB signal path gene NF-κB P65 in each group were detected. Data are mean ± SEM (*n* = 7). The letters ab are significant markers, and any difference with one of the same marker letters is not significant, and any difference with different marker letters is significant.

**Figure 6 nutrients-14-01069-f006:**
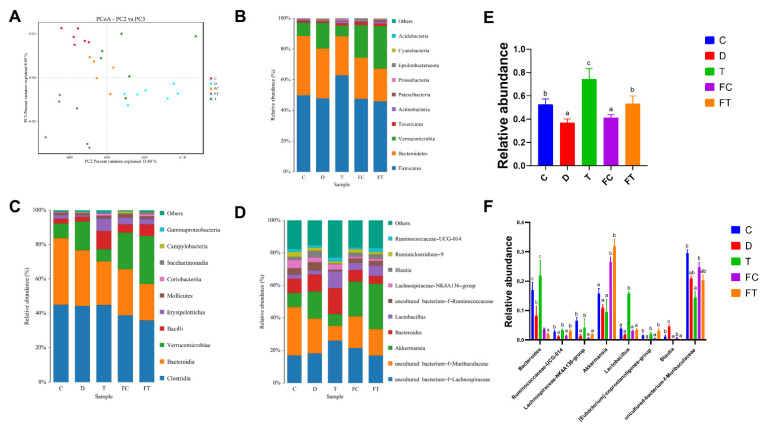
Taxifolin and FMT altered gut microbiota in DSS-induced intestinal mucositis model mice. (*n* = 7). (**A**) PCoA score based on weighted Unifrac metrics was different in each group; (**B**) Relative abundances of bacterial phyla level among five mice groups; (**C**) Relative abundances of bacterial class level among five mice groups; (**D**) Relative abundances of bacterial genus level among five mice groups; (**E**) the relative abundance of phylum; (**F**) the relative abundance of genus. The letters abcd are significant markers, and any difference with one of the same marker letters is not significant, and any difference with different marker letters is significant.

**Figure 7 nutrients-14-01069-f007:**
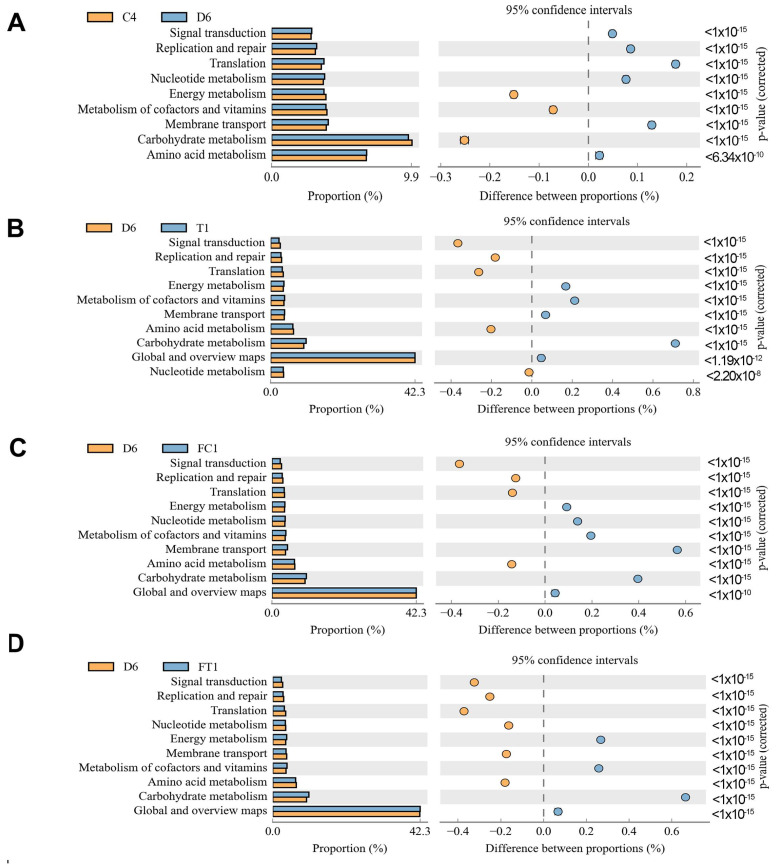
Predicted metabolic differences associated with different treatment in microbial communities found in the feces. Metabolic pathway was identified using PICRUSt software. (**A**) Difference with Control and DSS group in microbial communities; (**B**) Difference with DSS and Taxifolin group in microbial communities; (**C**) Difference with DSS and FC group in microbial communities; (**D**) Difference with DSS and FT group in microbial communities.

## Data Availability

Not applicable.

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
