# Peer review of "Taxifolin Alleviates DSS-Induced Ulcerative Colitis by Acting on Gut Microbiome to Produce Butyric Acid"

_nutrients, 2022, doi:10.3390/nu14051069_

Round 1

Reviewer 1 Report

  1. DSS treatment damaged crypt-villus structures and increased inflammatory cell (mainly lymphocytes) infiltration (Figure 2A). How do the authors determine that these inflammatory cells are lymphocytes?
  2. This study found that taxifolin regulated the SCFAs, inhibited the expression of TNF-α, IL-1β, and IL-6 in the colon, and alleviated DSS-induced colitis by regulating the gut microbiome in mice. However, some results are a bit overinterpreted:
  • The treatment effect of DSS-induced colitis in FMT (DSS + Taxifolin) group was better than taxifolin group (Figures 1D, 2, 3B). However, FMT (DSS + Taxifolin) treatment did not increase butyric acid and total acids (Figures 3C and 4). In addition, the anti-inflammatory effect of FMT (DSS + Taxifolin) treatment was not better than taxifolin alone (Figures 4 and 5). Whether FMT just restore the colonic mucosal damage and enhancing intestinal mucosal barrier by increase the Carbohydrate metabolism pathway. Could the authors explain the above phenomenon?
  • Please explain the effect of acetic acid producing bacteria on intestinal mucositis and gut microbiota and discuss on Figure 3C.
  • FMT (Control) treatment did not reverse DSS-induced IL-10 production. Why?
  1. In genially, the expression level of IkB cannot be used as an inflammatory marker. Moreover, it has no complementary relationship with NF-k Only the activity between them will affect each other. Therefore, IKKa/b would be better suited to illustrate the traditional NF-kB path.
  2. Although authors cited many references to illustrate taxifolin regulated intestinal flora indirectly affected the content of intestinal SCFAs, which reduced inflammation. However, no direct evidence showed that SCFAs inhibited inflammation in the DSS-treated intestinal mucositis. Authors should prove that.
  3. In some places, the authors may make a typo. Please correct it.
  • Page 6, Result 3.2, Line 233-235:In addition, the expression of GPR41 and GPR43 mRNA was “decreased” in FMT (DSS + Taxifolin) and FMT (Control) group compared with DSS group. ==> increased
  • Page 6, Result 3.2, Line 239-240:Pentatonic acid exhibited the same trend, but the difference did not reach significant (Figure “3B”). ==> 3C
  • Page 9, Result 3.6, Line 332:”Figureure7” shows…. ==> Figure 7
  1. Unit in Y axis of Figures should be revised:
  • Page 5 Figures 1B, 1D:body weight “(g)”;colon length “(cm)”
  • Page 6 Figures 3A, 3B:relative mRNA expression of xx in colon, “(pg/ml)” should be delete
  • Page 7 Figure 4 and 5:relative mRNA expression of xx in colon, “(pg/ml)” should be delete
  1. There is a severe plagiarism issue to raise. Although the methods or conditions of your laboratory's analysis platform remain unchanged, the author must still write in your way and not copy others' articles.

Author Response

Response to Reviewer Comments

  1. DSS treatment damaged crypt-villus structures and increased inflammatory cell (mainly lymphocytes) infiltration (Figure 2A). How do the authors determine that these inflammatory cells are lymphocytes?

Response: In the early stage of inflammation, inflammatory cell infiltration is mainly neutrophil infiltration, while in the late stage of inflammation, lymphocyte infiltration is the main infiltration. The section we selected in the late stage is not representative, and has been replaced with other sections, and words such as (mainly lymphocytes) have been deleted.

  1. This study found that taxifolin regulated the SCFAs, inhibited the expression of TNF-α, IL-1β, and IL-6 in the colon, and alleviated DSS-induced colitis by regulating the gut microbiome in mice. However, some results are a bit overinterpreted:
  • The treatment effect of DSS-induced colitis in FMT (DSS + Taxifolin) group was better than the taxifolin group (Figures 1D, 2, 3B). However, FMT (DSS + Taxifolin) treatment did not increase butyric acid and total acids (Figures 3C and 4).

Response:

Thanks for your question. The treatment effect of DSS-induced colitis in FMT (DSS + Taxifolin) group was better than the taxifolin group (Figures 1D, 2, 3B). However, these results were not significantly different between taxifolin group and FMT (DSS + Taxifolin) group. FMT (DSS + Taxifolin) The result of FMT is to prove that Taxifolin can indeed regulate the flora abundance of beneficial microorganisms in the gut microbiome. The SCFAs content in fecal in FMT (DSS + Taxifolin) and FMT (Control) group was significantly increased compared with DSS group. FMT (Control) group have a higher content in butyric acid, FMT (DSS + Taxifolin) group have a higher content in isobutyric acid and isovaleric acid compared with DSS group, which may be influenced by the abundance of Akkermansia and Lactobacillus and [Eubacterium]_coprostanoligenes_group in gut microbiota.

  • In addition, the anti-inflammatory effect of FMT (DSS + Taxifolin) treatment was not better than taxifolin alone (Figures 4 and 5). Whether FMT just restores the colonic mucosal damage and enhances intestinal mucosal barrier by increase the Carbohydrate metabolism pathway. Could the authors explain the above phenomenon?

Response: From the results (Figures 4 and 5), the two groups of fecal transplantation did alleviate DSS-induced intestinal mucositis compared with the DSS group, indicating that the intestinal flora actively participates in the pathological process of DSS-induced intestinal mucositis , emphasizing that intestinal flora plays an important role in intestinal mucositis, However, the effect of FMT (DSS+taxifolin) on intestinal mucositis and the abundance of intestinal microflora is not necessarily better than that of direct treatment with Taxifolin.

  • Please explain the effect of acetic acid producing bacteria on intestinal mucositis and gut microbiota and discuss on Figure 3C.

Response: Acetic acid producing bacteria is widely distributed within the bacterial spcies, Produced mainly by Akkermansia Muciniphila (A. muciniphila) and bifidobacteria in the gut. Intestinal acetic acid, propionic acid, butyric acid, etc. can increase the number and enhance the function of colon anti-inflammatory Treg cells in a GPR43-dependent manner. This protects the integrity of the gut and acts as an anti-inflammatory.

I have added the result analysis of acetic acid in Figure 3C.

  • FMT (Control) treatment did not reverse DSS-induced IL-10 production. Why?

Response: Thanks for your question,FMT(control) could not reverse the decrease of IL-10 induced by DSS, possibly because FMT treatment only reduced the decrease of IL-6, IL-1B and other pro-inflammatory factors, thus reducing the expression level of pro-inflammatory factors, so as to achieve the purpose of anti-inflammatory.

  1. In genially, the expression level of IkB cannot be used as an inflammatory marker. Moreover, it has no complementary relationship with NF-k Only the activity between them will affect each other. Therefore, IKKa/b would be better suited to illustrate the traditional NF-kB path.

Response: In response to your question, IKBα does not exist as an inflammatory marker in our test. IKBα binds to P65, thereby inhibiting the release of P65. After DSS induction, P65 is phosphorylated, and the inhibitory effect of IKBα is reduced, resulting in the release of P65.  Inflammation was induced, and after treatment with Taxifolin and FMT, the expression of IKBα was enhanced, which could inhibit the release of P65, thereby exerting its effect.

  1. Although authors cited many references to illustrate taxifolin regulated intestinal flora indirectly affected the content of intestinal SCFAs, which reduced inflammation. However, no direct evidence showed that SCFAs inhibited inflammation in the DSS-treated intestinal mucositis. Authors should prove that.

Response: Previous research proved that SCFAs could alleviate DSS-induced colitis by regulating autophagy via stabilizing HIF-1α. We have added the argument that SCFAs can alleviate DSS-induced enterocolitis in line 384 of the Discussion section.

  1. In some places, the authors may make a typo. Please correct it.
  • Page 6, Result 3.2, Line 233-235:In addition, the expression of GPR41 and GPR43 mRNA was “decreased” in FMT (DSS + Taxifolin) and FMT (Control) group compared with DSS group. ==> increased
  • Page 6, Result 3.2, Line 239-240:Pentatonic acid exhibited the same trend, but the difference did not reach significant (Figure “3B”). ==> 3C
  • Page 9, Result 3.6, Line 332:”Figureure7” shows…. ==> Figure 7

Response: Thanks for your notice, I have corrected these errors.

  1. Unit in Y axis of Figures should be revised:
  • Page 5 Figures 1B, 1D:body weight “(g)”;colon length “(cm)”
  • Page 6 Figures 3A, 3B:relative mRNA expression of xx in colon, “(pg/ml)” should be delete
  • Page 7 Figure 4 and 5:relative mRNA expression of xx in colon, “(pg/ml)” should be delete

Response: Thanks for your notice, I have revised these figures.

  1. There is a severe plagiarism issue to raise. Although the methods or conditions of your laboratory's analysis platform remain unchanged, the author must still write in your way and not copy others' articles.

Response: Thanks for your notice, I have described these experimental methods and conditions in my own words.

Reviewer 2 Report

Dear authors, it is difficult to understand what is or are the real new results of your experiments presented in the draft. Your paper is looking like others found on PubMed dealing with the same subject. The only point that can be considered as a new scientific knowledge could be the link between SCFA and GPR41 and 43 for which the information available are contradictory.

Please clarify the interest of FMT comparison.

I think your paper must reconsidered according the recent bibliography and must be focused on new real scientific information.

Author Response

Response to Reviewer Comments

Dear authors, it is difficult to understand what is or are the real new results of your experiments presented in the draft. Your paper is looking like others found on PubMed dealing with the same subject. The only point that can be considered as a new scientific knowledge could be the link between SCFA and GPR41 and 43 for which the information available are contradictory.

Please clarify the interest of FMT comparison.

I think your paper must reconsidered according the recent bibliography and must be focused on new real scientific information.

Response:

Dear reviewer, Thanks for your question, the innovation of our experiment is to discover the relationship between taxifolin and intestinal flora. Taxifolin can affect intestinal microbial metabolism by altering the abundance of gut microbes, thereby improving carbohydrate metabolism and ultimately exerting anti-inflammatory effects.

The innovation of our experiment is that Taxifolin can alleviate dss-induced colitis in mice, mainly by changing the abundance of gut microbes and affecting the metabolic level of gut microbes, especially carbohydrate metabolism, through SCFAs, to explain the anti-inflammatory mechanism of Taxifolin, thus providing new insights and ideas for the biological function and therapeutic potential of Taxifolin in the treatment of IBD.

Reviewer 3 Report

The manuscript “Taxifolin alleviate DSS-induced ulcerative colitis by acting on gut microbiome to produce butyric acid” is interesting while the is a number of concerns that require to address as mentioned below:

         The Introduction section needs to be re-write and should be focused and crisp. Why the incidence for only Canada was included, it looks very random? The Paragraph on the UC animal model looks unnecessary and could be omitted.

         In the method section, the induction of colitis in mice needs to provide a reference study, as in most of the study colitis is induced using 2-3% of DSS in drinking water, while in this study author used intraperitoneal administration of DSS (5%) for 7 days to induce colitis. Again, the dose is not clear (DSS-5%), it is required to provide the amount of volume of 5%-DSS was injected.

         The fecal microbiota preparation: It is required to provide a detailed methodology for microbial preparation like the timing of stool collection, whether it is used as frozen stool sample or fresh, the amount of feces used to make the suspension, how much bacterial suspension was used (no quantitation provided), whether a stool sample was collected from a new batch of mice or the same which used for DSS & DSS + Taxifolin experiment. The stool was dissolved in the water, which is inappropriate to make the microbial suspension, it needs to be dissolved in anaerobic PBS and in strictly anaerobic conditions. Whether the whole suspension (large sediments) was used to administer or only microbial suspension after centrifugation?

         It is advisable to give unique nomenclature to the group for microbial transplantation experiments.

         Using gene expression it is inconvincible to say the mechanism is controlled by the NF-κB signaling pathway, NF-κB signaling pathway is a multifactorial molecule and positively related to inflammation. It is required more experiment in vitro conditions to confirm Taxifolin mechanism through NF-κB signaling pathway.

         The column graph of the relative frequency of microbial population at phylum and genus level is not representative of relative frequency as presented in the respective relative abundances figure. In Fig 6E, the Verrucomicrobia in DSS group is identical/decreased as compared to control group and increased in DSS+ Taxifolin group as shown in column graph while it is opposite in abundance graph (Fig  6B), likewise Fig 6F is not representative of Fig 6D.

         Metabolic differences associated with different treatments should be analyzed together comparing all groups to the control group.

Author Response

 Response to Reviewer Comments

  1. The Introduction section needs to be re-write and should be focused and crisp. Why the incidence for only Canada was included, it looks very random? The Paragraph on the UC animal model looks unnecessary and could be omitted.

 Response: Thanks for your suggestion. The introduction in the article has been revised.

  1. In the method section, the induction of colitis in mice needs to provide a reference study, as in most of the study colitis is induced using 2-3% of DSS in drinking water, while in this study author used intraperitoneal administration of DSS (5%) for 7 days to induce colitis. Again, the dose is not clear (DSS-5%), it is required to provide the amount of volume of 5%-DSS was injected.

 Response: In the previous experiment, we used 3% DSS to induce colitis, but the effect was not ideal. After that, we checked relevant article and learned from other people's experimental methods. Experimental colitis was induced by replacing the drinking water with 5% DSS for 7days (from day 1 to day 7). I have revised the experimental method description.

  1. The fecal microbiota preparation: It is required to provide a detailed methodology for microbial preparation like the timing of stool collection, whether it is used as frozen stool sample or fresh, the amount of feces used to make the suspension, how much bacterial suspension was used (no quantitation provided), whether a stool sample was collected from a new batch of mice or the same which used for DSS & DSS + Taxifolin experiment. The stool was dissolved in the water, which is inappropriate to make the microbial suspension, it needs to be dissolved in anaerobic PBS and in strictly anaerobic conditions. Whether the whole suspension (large sediments) was used to administer or only microbial suspension after centrifugation?(重写实验方法部分)

 Response: Thanks for your suggestion, I have modified the description.

Fecal microbiota transplantation

For FMT experiment, 14 mice were randomly divided two groups: FMT-Control(FC) group and FMT-Taxifolin(FT)group (n=7/group). Experimental colitis was induced by replacing the drinking water with 5% DSS for 7 days (from day 1 to day 7). while mice in the control group were intraperitoneally given ddH2O. Fresh feces for murine FMT were shipped on ice to an anaerobic incubator in the laboratory within 30 minutes of defecation, and 200 mg of mixed feces were diluted with 2 ml of sterile saline and homogenized in a blender. The homogenate is then passed through a filter to remove small undigested particles from the fecal suspension, and 150ul (2×108-2×109 cfu/ml) bacterial suspension was used for gavage into each mouse.

FC mice were orally gavaged with fecal suspension from the control group once a day for 7 days (days 8-14). Meanwhile, mice in the FT group were orally gavaged with the fecal suspension of the DSS + paclitaxel group once a day for 7 days. The mice were not pretreated with antibiotics considering that antibiotic pretreatment has an effect on the efficacy of fecal transplantation [31] and may affect intestinal mucositis [32]. The body weight and diarrhea of the mice were recorded daily. At the end of the experiment, mice were executed by vertebral dislocation and the length of the induced colon was measured.

  1. It is advisable to give unique nomenclature to the group for microbial transplantation experiments.

Response: Thanks for your suggestion, give unique nomenclature to the group for microbial transplantation experiments.

FMT (DSS + Taxifolin) ==> FT

  1. Using gene expression it is inconvincible to say the mechanism is controlled by the NF-κB signaling pathway, NF-κB signaling pathway is a multifactorial molecule and positively related to inflammation. It is required more experiment in vitro conditions to confirm Taxifolin mechanism through NF-κB signaling pathway.

Response: In response to your question, our qPCR test is not intended to show that Taxifolin acts through the NF-KB signaling pathway. Previous studies by our research group have shown that Taxifolin acts through the NF-KB signaling pathway through Western Blot experiments (Dietary Taxifolin Protects Against Dextran Sulfate Sodium-Induced Colitis via NF-κB Signaling, Enhancing Intestinal Barrier and Modulating Gut Microbiota), Our qPCR assay is only a validation of findings.

  1. The column graph of the relative frequency of microbial population at phylum and genus level is not representative of relative frequency as presented in the respective relative abundances figure. In Fig 6E, the Verrucomicrobia in DSS group is identical/decreased as compared to control group and increased in DSS+ Taxifolin group as shown in column graph while it is opposite in abundance graph (Fig 6B), likewise Fig 6F is not representative of Fig 6D.

    Response: Thanks for your question, There was an error when uploading the relevant experimental data pictures, which has been modified in the article.

  1. Metabolic differences associated with different treatments should be analyzed together comparing all groups to the control group.

Response: In response to your question, the metabolic difference analysis requires a comparative analysis with the control group and other groups.  Our interpretation is that we predict the differences in microbial metabolism between each group and the DSS group by pairwise comparison of Picrust2 function, and this comparative analysis can prove that the control group, the taxifolin group, the FMT (control) group, the FMT (Taxifolin group), the FMT (Taxifolin group) group had higher carbohydrate metabolism levels than the DSS group.  It shows that the abundance of bacteria related to carbohydrate metabolism decreased after DSS treatment, while the abundance of bacteria related to carbohydrate metabolism was reversed after treatment with Taxifolin, FMT (control) or FMT (Taxifolin), indicating that Taxifolin and FMT have the function of regulating microbial metabolism.

Round 2

Reviewer 2 Report

Dear authors thanks for the modifications. Generally FT and FC have been substituted everywhere.

Part 3.1: is this part the confirmation that your model is compliant with already published articles on the same subject ? Is this part presenting as a validation of your model ?

3.2: You must be more rigorous Base don your results it is difficult to say that taxifolin reverse the situation while the quantity and the quality of the SCFA is very different in each group ?

3.3: FC reduced the inflammatory cytokines, but FMT alone also mediated the intestinal microflora with or without taxifolin. So what is the effect of Taxifolin alone in comparison with FMT alone ? This part is dedicated to inflammatory cytokines and you concluded the flora is mediated by taxifolin and then alleviates colitis. how can you conclude on flora only with cytokine assessment ?

3.4 same comments between the action of FMT alone and Taxifolin alone. To be clarified.

3.5: you write that FMt repairs in different way. Ok explain and modify please your conclusion.

4: discussion

437-438 you can wrote it while citing the ref 27 that demonstrated the gut flora modulation. This publication also demonstrated the improvement of the gut barrier.In all the discussion it is not clear how you demonstrated that Taxifolin is modulating the gut microflora. It is not clear is Taxifolin is acting directly on the flora or indirectly through its anti-inflammatory effect, that has as consequence the improvement of the flora balance. In your model it is explain that because of the inflammatory activity of DSS on gut tissues, the flora is in consequence modified in reaction of the inflammation.

What do you mean by Taxifolin prevent the development of intestinal mucosa ?

You concluded that Taxifolin improve the intestinal barrier. So where are the description of the method and the experiment assessing the barrier ?

At the you conclude that TAxifolin is a treatment for diarrhea. I'm not sure, because diarrhea may have several origins, diarrhea associated colitis could be more compliant with your model.

Author Response

Response to Reviewer Comments

  1. 3.1: is this part the confirmation that your model is compliant with already published articles on the same subject ? Is this part presenting as a validation of your model ?

Response: Thanks for your question. Part 3.1 is to verify the success of modeling on the one hand, and to show phenotypically that Citicoline can relieve colitis. In the process of modeling, we did several indexes such as blood in stool, diarrhea, and body weight to verify whether the modeling was successful. Specifically in the context of this experiment, we are mainly exploring colitis, and the diarrhea index does not indicate the success of colitis modeling, so we now choose the DAI index to illustrate the success of modeling.

  1. 2: You must be more rigorous Base don your results it is difficult to say that taxifolin reverse the situation while the quantity and the quality of the SCFA is very different in each group ?

Response: Thanks for your question. The number of SCFAs may indeed be different in different groups because we explored whether the number of SCFAs produced was the same from three angles. the FC group was to investigate the effect on the number of SCFAs after the action of intestinal flora in normal rats, the FT group was to investigate the effect on the number of SCFAs after the improvement of intestinal flora in the T group after DSS induction, and the T group was to investigate the effect of citicoline directly effect on the number of SCFAs. However, their effects on the number of SCFAs were different, and their effects on the specific types of SCFAs were also different.

  1. 3: FC reduced the inflammatory cytokines, but FMT alone also mediated the intestinal microflora with or without taxifolin. So what is the effect of Taxifolin alone in comparison with FMT alone ? This part is dedicated to inflammatory cytokines and you concluded the flora is mediated by taxifolin and then alleviates colitis. how can you conclude on flora only with cytokine assessment ?

Response: Thanks for your question. We are not comparing the Taxifolin group with the FMT group alone. The Taxifolin group, the FC group, and the FT group explain the relief of colitis from three aspects. The significance of the FC group is to explain the intestinal flora of normal mice.  Colitis can be relieved, and the significance of the FT group is that after treatment with Taxifolin, the intestinal flora destroyed by Dss can be improved, thereby relieving colitis.  The results of the T group indicated that Taxifolin had an effect on inflammatory factors.  The results of the FT group indicated that Taxifolin could improve the gut microbiota induced by DSS, thereby affecting inflammatory factors, which in turn linked inflammatory factors to the intestinal microbiota.

  1. 4 same comments between the action of FMT alone and Taxifolin alone. To be clarified.

Response: Thanks for your question. Fecal transplantation in both groups did reduce DSS-induced intestinal mucositis compared to the DSS group, suggesting that intestinal flora actively participates in the pathological process of DSS-induced intestinal mucositis and emphasizing the important role of intestinal flora in intestinal mucositis, however, the effect of FMT (DSS+Taxifolin) on intestinal mucositis and intestinal microflora abundance was not necessarily superior to direct Taxifolin treatment.

  1. 5: you write that FMt repairs in different way. Ok explain and modify please your conclusion.

Response: Thanks for your suggestion. I have modified my description in the results.

  1. 437-438 you can wrote it while citing the ref 27 that demonstrated the gut flora modulation. This publication also demonstrated the improvement of the gut barrier.In all the discussion it is not clear how you demonstrated that Taxifolin is modulating the gut microflora. It is not clear is Taxifolin is acting directly on the flora or indirectly through its anti-inflammatory effect, that has as consequence the improvement of the flora balance. In your model it is explain that because of the inflammatory activity of DSS on gut tissues, the flora is in consequence modified in reaction of the inflammation.

Response: Thanks for your question. First, we used DSS-induced colitis to reduce the abundance of beneficial flora in the intestinal flora, and transplanting the intestinal gut flora of Taxifolin+DSS into the mice of the enteritis model improved the abundance of intestinal flora and increased the abundance of beneficial bacteria, indicating that it was indeed Taxifolin that improved the disrupted intestinal flora from the FMT (Taxifolin+DSS) this From the results of FMT(Taxifolin+DSS) group, it is true that Taxifolin first changed the balance of intestinal flora and then improved the intestinal inflammation.

  1. What do you mean by Taxifolin prevent the development of intestinal mucosa ?

Response: Thank you for your question. I made a mistake in my statement. What I was trying to convey was that Taxifolin can relieve intestinal inflammation and stop the progression of colitis. I have revised my statement in the article.

  1. You concluded that Taxifolin improve the intestinal barrier. So where are the description of the method and the experiment assessing the barrier ?

Response: Thank you for your question. Our HE staining results as well as the histological damage score can only be speculated to improv e the intestinal barrier, and I have deleted the intestinal mucosal barrier related description in the conclusion.

  1. At the you conclude that Taxifolin is a treatment for diarrhea. I'm not sure, because diarrhea may have several origins, diarrhea associated colitis could be more compliant with your model.

Response: Thank you for your suggestion. Our experiments were performed with a model of DSS-induced colitis, a model of inflammatory bowel disease that cannot be over-interpreted to be associated with diarrhea. We have modified this argument in the conclusion section.

Reviewer 3 Report

  • The method says the control group received intraperitoneal injection of ddH2. If the DSS group received DSS in drinking water, then why control group has received the intraperitoneal injection of ddH2O? (Line 182-183; 192-193)
  • Line 193-195; “Fresh feces for murine FMT were shipped on ice to an anaerobic incubator in the laboratory within 30 minutes of defecation, and 200 mg of mixed feces were diluted with 2 ml of sterile saline and homogenized in a blender”. This approach is not correct, as homogenization using a blender may cause bacterial lysis and death.

  • Line 201: It is written DSS + paclitaxel instead of DSS+ Taxifolin.

  • Need to provide primers detail used for qRT.

  • Line 419-420 need to be re-write.

  • In 3.6 Predicted metabolomic profiles of microbiota from Control and DSS and Taxifolin and FMT group, the author says that FMT may restore the colonic mucosal damage by increasing the carbohydrate metabolism pathway, while in the figure FMT group is similar as control, DSS + Taxifolin and FC group.

  • All the figures should have a number of biological replicates applied for the specific experiment and which statistical analysis was applied.

  • All the short forms should also be explained in the figure legends.

Author Response

 Response to Reviewer Comments

  1. The method says the control group received intraperitoneal injection of ddH2. If the DSS group received DSS in drinking water, then why control group has received the intraperitoneal injection of ddH2O? (Line 182-183; 192-193)

 Response: Thanks for your suggestion. I made a mistake in my statement. I have revised my statement in the article.

  1. Line 193-195; “Fresh feces for murine FMT were shipped on ice to an anaerobic incubator in the laboratory within 30 minutes of defecation, and 200 mg of mixed feces were diluted with 2 ml of sterile saline and homogenized in a blender”. This approach is not correct, as homogenization using a blender may cause bacterial lysis and death.

 Response: Thanks for your suggestion. My expression is not accurate, it should be Using a homogenizer, gently mix the fecal bacteria solution. I have revised my statement in the article.

  1. Line 201: It is written DSS + paclitaxel instead of DSS+ Taxifolin.

 Response: Thanks for your notice. I made a mistake in my statement. I have revised my statement in the article.

  1. Need to provide primers detail used for qRT.

Response: Thanks for your notice. I have added primer design details in Supplement table 1.

  1. Line 419-420 need to be re-write.

Response: Thanks for your suggestion. I have revised my statement in the article.

  1. In 3.6 Predicted metabolomic profiles of microbiota from Control and DSS and Taxifolin and FMT group, the author says that FMT may restore the colonic mucosal damage by increasing the carbohydrate metabolism pathway, while in the figure FMT group is similar as control, DSS + Taxifolin and FC group.

    Response: Thanks for your question. In a single figure, The proportion of abundance of different functions in two samples or two groups of samples is shown in the left panel of the image, and the proportion of differences in the abundance of functions in the 95% confidence interval is shown in the middle, and the rightmost value is the p-value. So there is a significant difference in the proportion of functional differences in carbohydrate metabolism.

  1. All the figures should have a number of biological replicates applied for the specific experiment and which statistical analysis was applied.

Response: Thanks for your suggestion. I have revised my statement in the article.

  1. All the short forms should also be explained in the figure legends.

Response: Thanks for your suggestion. I have revised my statement in the article.